# Evaluating Bezlotoxumab-Fidaxomicin Combination Therapy in Clostridioides Infection: A Single-Center Retrospective Study from Aichi Prefecture, Japan

**DOI:** 10.3390/antibiotics14030228

**Published:** 2025-02-24

**Authors:** Jun Hirai, Nobuaki Mori, Yuki Hanai, Nobuhiro Asai, Mao Hagihara, Hiroshige Mikamo

**Affiliations:** 1Department of Clinical Infectious Diseases, Aichi Medical University Hospital, Aichi 480-1195, Japan; hiraichimed@gmail.com (J.H.); forest.catch22@gmail.com (N.M.); nobuhiro0204@gmail.com (N.A.); 2Department of Infection Control and Prevention, Aichi Medical University Hospital, Aichi 480-1195, Japan; 3Division of Infection Control and Prevention, Nippon Medical School Chiba Hokusoh Hospital, Chiba 270-1694, Japan; 4Department of Clinical Pharmacy, Faculty of Pharmaceutical Sciences, Toho University, Chiba 274-8510, Japan; yuki.hanai@phar.toho-u.ac.jp; 5Department of Molecular Epidemiology and Biomedical Sciences, Aichi Medical University Hospital, Aichi 480-1195, Japan; hagimao@aichi-med-u.ac.jp

**Keywords:** bezlotoxumab, fidaxomicin, combination therapy, *Clostridioides difficile*, recurrence, severe diarrhea

## Abstract

**Background/Objectives:** *Clostridioides difficile* infection (CDI) poses a significant healthcare challenge, with recurrence rates reaching 30%, leading to substantial morbidity and costs. Fidaxomicin (FDX) and bezlotoxumab (BEZ) have shown potential in reducing recurrence; however, real-world data on the efficacy of their combination in high-risk CDI patients remain limited. This study aimed to evaluate the efficacy and safety of FDX + BEZ compared with FDX alone in CDI patients with recurrence risk factors. **Methods:** CDI patients with ≥two recurrence risk factors treated with FDX alone or FDX + BEZ were analyzed. Sixteen factors were evaluated as risk factors for recurrent CDI based on findings from previous studies. Patients with FDX treatment duration <10 days or other CDI treatment prior to FDX were excluded. Outcomes included recurrence within 2 months, global and clinical cure rates, and adverse events. Univariate and multivariate analyses were performed to evaluate efficacy. **Results:** Among 82 patients, the FDX + BEZ group (*n* = 30) demonstrated significantly higher global (86.7% vs. 65.4%; *p* < 0.05) and clinical cure rates (90.0% vs. 69.2%; *p* < 0.05) compared with the FDX-alone group (*n* = 52), despite more severe cases in the combination group. Recurrence rates were non-significantly lower in the FDX + BEZ group (3.3% vs. 11.5%). Combination therapy also accelerated diarrhea resolution without additional adverse events. Multivariate analysis identified FDX + BEZ as significantly associated with improved clinical cure (adjusted odds ratio 4.167; 95% CI: 1.029–16.885). **Conclusions:** FDX + BEZ therapy offers superior efficacy and safety in CDI patients with recurrence risk factors, presenting a promising strategy for optimizing CDI management.

## 1. Introduction

*Clostridioides difficile* infection (CDI) is a significant healthcare-associated infection worldwide [1,2]. Recurrence rates of CDI remain high at up to 30% despite treatment with standard antibiotics, such as vancomycin, leading to increased morbidity, mortality, and healthcare costs [3,4]. Novel treatment approaches, such as bezlotoxumab (BEZ) and fidaxomicin (FDX), have shown promise in reducing recurrence rates in clinical trials [5,6]. Bezlotoxumab, a humanized immunoglobulin G1 monoclonal antibody, targets *Clostridioides difficile* (CD) toxin B, neutralizes the toxin, and prevents damage to colon cells, resulting in reduced recurrence rates when administered in combination with a standard antibiotic, such as vancomycin [7,8]. Fidaxomicin, a narrow-spectrum antibiotic, has been used to effectively treat patients with CDI with recurrence risk factors and those with initial CDI episodes [6]. Compared with vancomycin and metronidazole, FDX has been shown to have a significantly lower recurrence rate of CDI [9,10,11,12].

Several reports of clinical trials exist on the actual recurrence rate for the combination therapy of BEZ and anti-CDI drugs [13,14,15]; however, there are no real-world data on the efficacy and safety of the combination therapy of BEZ and FDX compared with FDX alone in clinical settings. The MODIFY I and MODIFY II trials are global, randomized, double-blind, placebo-controlled clinical trials conducted to assess the efficacy and safety of BEZ for preventing CDI recurrence in adults. However, in these trials, the risk factors for CDI relapse were not assessed before BEZ administration, and 40.5% of the cases treated with FDX + BEZ were relapsed cases, not initial CDI cases. Furthermore, data on the relapse-suppressing effect of BEZ were mainly obtained from patients who used vancomycin and metronidazole. Therefore, a sufficient analysis of the treatment data, including those on the prevention of CDI recurrence following FDX + BEZ therapy in primary CDI patients with relapse risk factors, could not be performed [5]. Consequently, in real-world clinical practice, when managing a patient with a high risk of relapse, clinicians must consider whether the combined FDX and BEZ therapy is more effective compared with FDX monotherapy.

The objective of this retrospective single-center study conducted in Aichi, Japan, was to evaluate the effectiveness and safety of combination therapy with BEZ and FDX compared with FDX monotherapy in CDI patients with recurrence risk factors.

## 2. Results

During the study period, 192 patients were diagnosed with CDI. Of these, 55 and 30 were assigned to the FDX and FDX + BEZ groups, respectively. In the FDX cohort, three patients whose treatment with FDX lasted for <10 days due to deaths resulting from other causes than CDI were excluded. Therefore, 82 patients were included in the present study (Figure 1). Demographic and baseline characteristics of the study population are presented in Table 1. The two groups were well matched in terms of baseline characteristics such as age, sex, body mass index, community-onset ratio, admission to the intensive care unit (ICU) at the time of CDI diagnosis, medical history of CDI, and past hospitalization. Patient condition at the time of CDI diagnosis, in terms of body temperature (axillary temperature), complaint of abdominal pain, number of diarrhea per day, and bloody stool, were similar in both groups. Intergroup differences in fasting, use of enteral feeding, hospitalization for more than 1 month before CDI onset, a history of abdominal surgery, and use of antimicrobials for at least 3 days after CDI diagnosis were not significant. Baseline comorbidities, such as diabetes mellitus, were also not significantly different, except for chronic kidney disease (CKD). Regarding antibiotics administration at CDI onset, differences between groups in the types of antimicrobials used were not significant. The ratio of concomitant medication use, such as potassium-competitive acid blockers, immunosuppressants, anti-cancer chemotherapy, and probiotics before/after CDI diagnosis, also did not differ significantly between the two groups. In laboratory data, the number of patients with a creatinine level > 1.5 mg/dL was significantly higher in the FDX + BEZ group compared with the FDX group (*p* < 0.01). The proportions of severe cases based on the Infectious Diseases Society of America (IDSA)/Society for Healthcare Epidemiology of America (SHEA) criteria were 46.2% and 76.7% in the FDX and FDX + BEZ groups, respectively, with significantly more severe cases in the FDX + BEZ group (*p* < 0.01). The ratio of severe cases based on the Mikamo–Nakamura (MN) criteria was also significantly higher in the FDX + BEZ group compared with the FDX group (*p* < 0.01). Probiotic usage before and after CDI diagnosis was not significantly different between the two groups (*p* = 0.104, *p* = 0.47, respectively). Regarding the secondary endpoints, no adverse events were observed in either group. In addition, the two groups showed no significant difference in 30-day all-cause mortality following CDI treatment (*p* = 0.2770) during the study period (two patients treated with FDX alone died of cancer).

In the univariate analysis, no significant difference in the CDI recurrence rate was observed between the two cohorts (FDX + BEZ in 11.5% vs. FDX alone in 3.3%, *p* = 0.20). However, the FDX + BEZ group showed a significant improvement in global cure (65.4% vs. 86.7%, *p* < 0.05) and clinical cure (69.2% vs. 90.0%, *p* < 0.05) compared with the FDX group. In this study, BEZ was administered an average of 5.4 days after the start of CDI treatment, and of 30 patients, the 24 (80%) who received BEZ experienced a resolution of diarrhea (types 3–4 on the Bristol Stool Scale) within 2 days of administration of BEZ. A subgroup analysis was conducted to assess the potential impact of BEZ administration timing on treatment outcomes. Patients who received BEZ within 5 days of FDX initiation (*n* = 13) were compared to those who received BEZ after 5 days (*n* = 17). No statistically significant differences were found between the two groups for clinical cure (*p* = 0.390), global cure (*p* = 0.773), or recurrence rate (*p* = 0.374)). We compared patients who achieved clinical cure with those who did not, and the variables associated with clinical cure at *p* < 0.05 in univariate analysis were included in multivariate logistic regression analysis. As shown in Table 2, FDX + BEZ therapy and the use of probiotics after CDI diagnosis were significantly associated with clinical cure (adjusted odds ratio [AOR]: 4.167, 95% confidence interval [CI]: 1.029–16.885, *p* = 0.046, and AOR: 3.403, 95% CI: 1.045–11.079, *p* = 0.042, respectively). In contrast, steroid use was significantly associated with clinical cure failure (AOR: 0.236, 95% CI: 0.068–0.815, *p* = 0.022). Even in the severe cases defined by IDSA/SHEA guidelines for CDI, those in the FDX + BEZ group had significantly higher global and clinical cure rates than did those in the FDX group (*p* = 0.02 and *p* < 0.01, respectively), although multivariate analysis could not be performed due to the small number of clinical failure cases (Table 3).

## 3. Discussion

This study was retrospective and non-randomized, meaning that treatment allocation was based on clinical judgment rather than systematic randomization. While this approach reflects real-world clinical practice, it introduces potential selection bias. In this study, we evaluated the efficacy and safety of FDX + BEZ compared with FDX monotherapy in CDI patients with recurrence risk factors. A notable finding of our research was that the FDX + BEZ group had significantly higher clinical and global cure rates than the FDX-alone group did, even in the subset analysis of severe cases. This suggests that this combination therapy offers an enhanced therapeutic approach for high-risk patients with CDI. The observed recurrence rates in the FDX + BEZ group were also lower than those reported in previous studies of combinations of BEZ with vancomycin or metronidazole [16,17,18,19]. Thus, this highlights the potential synergistic effect of FDX and BEZ and underscores the efficacy of BEZ in high-risk populations, particularly in those with severe CDI or with multiple recurrence risk factors. The combination of FDX and BEZ represents a promising therapeutic strategy for reducing CDI recurrence. FDX, a narrow-spectrum macrocyclic antibiotic, selectively inhibits *C. difficile* RNA polymerase, reducing bacterial replication while preserving the gut microbiota. Unlike vancomycin, FDX minimally disrupts commensal anaerobes, which are crucial in preventing CDI recurrence [20]. On the other hand, BEZ is a monoclonal antibody that neutralizes *C. difficile* toxin B, thereby preventing its cytotoxic effects on intestinal epithelial cells [21]. Since BEZ does not have direct antibacterial properties, it does not affect bacterial burden but instead mitigates toxin-mediated damage and inflammation [21]. We assume that the potential synergy between these two agents arises from their complementary mechanisms: FDX reduces bacterial burden while BEZ prevents mucosal injury and recurrence by neutralizing toxin B. This dual-action approach would be particularly beneficial in patients at high risk for recurrence, as it simultaneously addresses both pathogen reduction and toxin-mediated pathogenesis. Further studies, including in vitro and in vivo models, are warranted to explore the precise molecular interactions and clinical implications of this combination therapy. The multivariable logistic regression analysis further supported the role of combination therapy, showing a significant association between the use of FDX + BEZ and clinical cure. In addition, no serious adverse events were observed in the FDX + BEZ group in this study, suggesting that these treatments may be safe. These safety findings may help expand treatment options, particularly for patients at high risk of relapse.

Another finding was that 80% of patients in the FDX + BEZ group experienced a resolution of diarrhea within 2 days of BEZ administration. This suggests the need to administer BEZ as early as possible after CDI diagnosis, although current guidelines do not specify the appropriate timing of BEZ administration [12,22,23]. The optimal timing of BEZ administration in the treatment of CDI remains a topic of clinical interest. In our study, we observed no significant difference in clinical outcomes based on whether BEZ was administered within or after 5 days of FDX initiation. Given our sample size limitations, larger prospective studies are needed to definitively determine the optimal timing of BEZ in the management of CDI. This rapid symptomatic relief highlights an important clinical benefit of BEZ in combination with FDX, particularly in patients with severe CDI or those at high risk of relapse. BEZ provides passive immunity to CD toxin B by directly binding to regions of the combined repetitive oligopeptide domains of toxin B, further blocking the toxin’s receptor-binding pockets [24,25], which, as we hypothesized, may induce earlier resolution of diarrhea. Once bound, toxin B is neutralized, preventing toxin B-mediated inflammation and damage to colon cells and subsequent symptoms of CDI [24,25]. Previous pivotal studies, such as the MODIFY trials, demonstrated the efficacy of BEZ in preventing recurrent CDI but did not specifically evaluate its acute symptomatic relief effect [12,22,25]. Based on our findings, although the number of cases studied is small, the early administration of BEZ is recommended for patients with CDI who are at high risk of recurrence and who experience frequent or severe diarrhea. Future studies should aim to further elucidate the timing optimization of BEZ administration to maximize its clinical benefit.

The guidelines of each country recommend BEZ for patients with CDI who are aged ≤ 65 years, have severe CDI, have an immunocompromised status, or have had any CDI episode with at least one additional risk factor for recurrence [12,23,26]. In this study, we used BEZ in combination with FDX for patients who had at least two of the recurrence risk factors listed in Table 2, achieving a lower recurrence rate than that reported in previous studies. However, due to the small number of relapse cases, we could not determine which specific recurrent risk factors make BEZ the most effective. Because BEZ is an expensive drug, patients who receive BEZ must be carefully selected from those with multiple risk factors for relapse. In the future, it will be necessary to identify the patient groups that benefit most from the combined use of BEZ and FDX, both in terms of clinical efficacy and cost-effectiveness.

In recent years, it has become clear that the development of CDI is associated with a decrease in gut flora diversity, highlighting the need for treatment strategies that minimize disruption of the gut microbiota while preventing recurrence [27,28]. FDX has a narrower spectrum of activity compared with metronidazole or vancomycin and has been reported to have less effect on gut flora [29]. Two randomized controlled trials comparing the therapeutic efficacy of FDX and vancomycin as first-line CDI therapy [22] showed that the cure rate of FDX was equivalent to that of vancomycin; however, the recurrence rate was significantly lower with FDX. These results highlight the importance of FDX in the treatment of CDI, not only for its therapeutic efficacy but also for its role in preventing recurrence and its alignment with antimicrobial stewardship principles. Thus, FDX is recommended as first-line therapy in the IDSA/SHEA [23] and European Society of Clinical Microbiology and Infectious Diseases guidelines [29,30,31]. In this study, no serious adverse events were observed in the FDX + BEZ group, and a higher clinical cure rate was achieved with FDX + BEZ than with FDX alone, in addition to the earlier resolution of diarrhea. This suggests a synergistic effect that improves the recovery of CDI patients, with less disruption of gut flora diversity. Therefore, combination therapy with FDX and BEZ would be a promising treatment option, especially for CDI patients with a high risk of recurrence and frequent diarrhea.

In this study, the use of probiotics in combination with CDI treatment was associated with clinical cure; however, there is insufficient evidence on the efficacy of probiotics in the treatment of CDI [30,31,32]. Moreover, there are safety concerns, such as bacteremia, particularly in severely ill or immunosuppressed patients [33]. Therefore, the use of probiotics in the treatment of CDI is not recommended in major guidelines worldwide [12,23,26]. In addition, although the use of steroids has been reported to be a risk factor for the recurrence of CDI [34], we could not identify studies related to clinical cure failure as in this study. Based on these findings, the lack of robust evidence requires further investigation through well-designed studies.

This study has certain limitations. First, it was a retrospective single-center study with a small sample size; therefore, the results should be interpreted with caution and require validation in future studies. However, no studies have compared the clinical effectiveness of FDX versus FDX + BEZ in CDI patients with a high risk for recurrence. Therefore, this study may help clinical physicians make informed decisions regarding the management of CDI in patients with recurrence risk factors. Secondly, this study could not determine which patients with relapse risk factors would benefit most from the addition of BEZ to FDX, which may be due to the small sample size. In the future, it will be essential to identify specific recurrence risk factors or combinations of risk factors that warrant treatment with BEZ alongside FDX. Third, in this study, BEZ was administered, on average, 5.4 days after the initiation of FDX; however, it is still unclear whether this timing affects the clinical and overall cure rates. Therefore, when treating CDI patients with FDX, the effect of the timing of BEZ administration on treatment efficacy needs to be investigated in the future. Fourth, we were unable to evaluate differences in the therapeutic efficacy of FDX vs. FDX + BEZ against hypervirulent strains, such as ribotype 027/078/244, which are rarely isolated in Japan [35], because ribotyping analysis was not performed in the present study.

In conclusion, FDX + BEZ combination therapy offers a promising treatment option for CDI patients at high risk of recurrence, demonstrating improved clinical outcomes even in severe cases. In addition, the present study showed that FDX + BEZ combination therapy also resolved diarrhea faster than FDX alone in CDI patients. Future research should focus on larger studies to validate these findings and optimize treatment strategies for this challenging infection.

## 4. Materials and Methods

This retrospective study was carried out at the Aichi Medical University Hospital, a tertiary care facility in Japan with 900 inpatient beds, including a 40-bed ICU. The study included symptomatic CDI patients aged over 18 years who had factors for recurrence and were treated with either FDX combined with BEZ or FDX alone for a 10-day initial course between August 2019 and April 2024. Their medical records were thoroughly reviewed for analysis. Patient allocation to the FDX monotherapy group or the FDX + BEZ combination therapy group was not randomized but instead based on clinical judgment. The attending physicians made treatment decisions based on individual patient factors, including disease severity, risk of recurrence, and comorbidities. Patients on FDX treatment with duration <10 days or on other CDI treatments (such as metronidazole and vancomycin) prior to FDX were excluded. The primary outcome was the difference between the two cohorts in the ratio of recurrence within 2 months after treatment, the global and clinical cure rates of CDI, and adverse events. The secondary outcome was all-cause mortality within 30 days.

CDI was diagnosed based on the following criteria [36]: (1) the occurrence of ≥3 loose stools (types 5–7 on the Bristol Stool Scale) within 24 h before stool sampling and (2) a positive result for CD toxins using a rapid immunoassay for glutamate dehydrogenase (GDH) and toxin detection (C. DIFF QUIK CHEK COMPLETE; TechLab, Blacksburg, VA, USA, or GE Test Immunochromato-CD GDH/TOX; Nissui Pharmaceutical Co., Ltd., Tokyo, Japan), or polymerase chain reaction targeting the toxin B gene using the Cepheid GeneXpert *C. difficile* Assay (Beckman Coulter Inc., Tokyo, Japan). The C. DIFF QUIK CHEK COMPLETE kit (TechLab) was employed from August 2019 to March 2022, while the GE Immunochromatography-CD GDH/TOX test was utilized from March 2023.

The following 16 factors were identified as relapse risk factors for recurrent CDI based on findings of previous studies [23,34,37,38,39,40,41,42,43,44,45,46]: (1) age ≥ 65 years; (2) use of antimicrobials for at least 3 days after CDI diagnosis; (3) CKD; (4) a history of CDI; (5) use of antacids (proton pump inhibitors, H2 blockers, or potassium-competitive acid blockers); (6) hospitalization within 3 months prior to CDI onset; (7, 8) use of fluoroquinolones or carbapenem within 1 month prior to CDI onset; (9, 10) presence of solid malignancy or hematologic cancer; (11) admission to the ICU at the time of CDI diagnosis; (12) history of cephalosporin usage prior to CDI diagnosis; (13) history of abdominal surgery; (14) steroid use; (15) presence of inflammatory bowel disease; and (16) severe CDI. CDI patients with two or more recurrence risk factors were administered either FDX only or FDX + BEZ.

The severity of CDI was assessed according to the criteria outlined in the SHEA/IDSA guidelines [47] and the MN scoring system [36,48]. According to the SHEA/IDSA guidelines, severe CDI is defined by a white blood cell count exceeding 15,000 cells/mL or a serum creatinine level of 1.5 mg/dL or higher [47]. The MN criteria, a Japanese scoring framework developed for CDI severity evaluation, utilize nine categorical variables: patient age, presence of abdominal pain and distension, body temperature, frequency of diarrhea, hematochezia, white blood cell count, estimated glomerular filtration rate (eGFR), serum albumin levels, and imaging results. Each variable is graded on a 3-point scale, with total scores stratified into four categories: ≤4 (mild), 5–9 (moderate), 10–13 (severe), and ≥14 (super-severe) [36].

Global cure was defined as achieving clinical cure without recurrence within 2 months following treatment completion [19]. The global cure rate was calculated as the proportion of patients who successfully completed treatment either on FDX + BEZ or FDX alone. Clinical cure was defined as the resolution of diarrhea based on the Bristol Stool scale consistency within 2 days after the completion of CDI therapy [46]. Recurrent CDI was defined as the need to reinitiate CDI treatment due to diarrhea accompanied by a confirmatory positive test within eight weeks of the initial CDI treatment episode [47].

Monitoring for side effects such as kidney damage, liver damage, and heart failure was performed during the CDI treatment and for 7 days after the CDI treatment.

Categorical data, including age, are presented as medians with interquartile ranges. Continuous variables were analyzed using the Mann–Whitney U-test, while categorical variables were compared using the χ^2^ test or Fisher’s exact test. Potentially significant variables identified in the univariate analysis (*p* < 0.05) were included in the multivariate logistic regression model to identify clinical cure-associated factors. In cases where multicollinearity was detected among the independent variables, only one variable from each correlated pair was included in the analysis. Statistical significance was defined as a *p*-value of less than 0.05. All statistical analyses were conducted using SPSS version 26 for Windows (IBM Corp., Armonk, NY, USA).

## Figures and Tables

**Figure 1 antibiotics-14-00228-f001:**
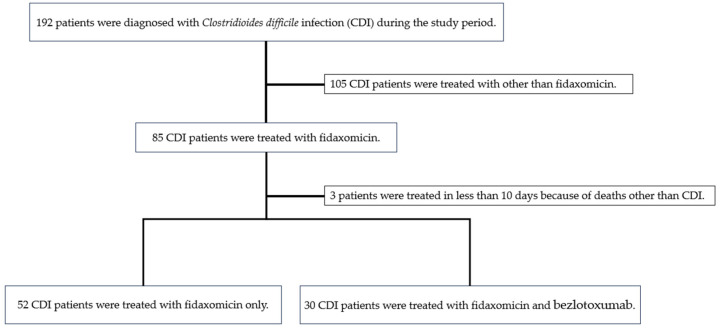
Flowchart of the inclusion and exclusion of *Clostridioides difficile* infection patients in the study cohort.

**Table 1 antibiotics-14-00228-t001:** Baseline characteristics between two cohorts.

Variables	FDX (*n* = 52)	FDX + BEZ (*n* = 30)	*p* Value
Age (years), median (IQR)	73 (32–93)	77 (30–90)	0.1560
Age ≥ 65 years, no. (%)	34 (65.4)	25 (83.3)	0.0810
Age ≥ 75 years, no. (%)	25 (48.1)	18 (60.0)	0.2980
Age ≥ 80 years, no. (%)	13 (25.0)	13 (43.3)	0.0860
Female sex, no. (%)	25 (48.1)	17 (56.7)	0.4540
Body mass index < 18.5, no. (%)	22 (42.3)	12 (40.0)	0.8380
Past medical history of CDI, no. (%)	4 (7.7)	3 (10.0)	0.7190
Past hospitalization within 3 months before onset of CDI, no. (%)	24 (46.2)	9 (30.0)	0.1510
Past hospitalization < 1 year, no. (%)	38 (73.1)	15 (50.0)	0.0350
Temperature > 37.8 °C at diagnosis of CDI, no. (%)	23 (44.2)	14 (46.7)	0.8310
Abdominal pain at diagnosis of CDI, no. (%)	23 (44.2)	16 (53.3)	0.4270
Number of diarrhea per day > 5 at diagnosis of CDI, no. (%)	22 (42.3)	17 (56.7)	0.2100
Bloody stool, no. (%)	5 (9.6)	2 (6.7)	0.6450
Fasting, no. (%)	7 (13.5)	2 (6.7)	0.3430
Enteral feeding, no. (%)	5 (9.6)	7 (23.3)	0.0900
Hospitalization for more than 1 month before onset of CDI, no. (%)	16 (30.8)	11 (36.7)	0.5840
Use of antimicrobials for at least 3 days after CDI diagnosis, no. (%)	23 (44.2)	11 (36.7)	0.5030
Comorbidities, no. (%)			
Diabetes mellitus	19 (36.5)	11 (36.7)	0.9910
Chronic kidney disease	19 (36.5)	23 (76.7)	<0.01
Hemodialysis	10 (19.2)	3 (10.0)	0.2700
Heart failure/ischemic heart disease	12 (23.1)	9 (30.0)	0.4890
Chronic liver disease	2 (3.8)	0 (0)	0.2770
Chronic obstructive pulmonary disease	4 (7.7)	2 (6.7)	0.8640
Cerebrovascular disease	6 (11.5)	5 (16.7)	0.5120
Inflammatory bowel disease	4 (7.7)	1 (3.3)	0.4270
Solid malignancy	16 (30.8)	9 (30.0)	0.9420
Hematologic malignancy	3 (5.8)	3 (10.0)	0.4790
History of abdominal surgery, no. (%)	15 (28.8)	4 (13.3)	0.1090
Community onset CDI, no. (%)	7 (13.5)	2 (6.7)	0.3430
Intensive care unit at diagnosis of CDI, no. (%)	3 (5.8)	2 (6.7)	0.8700
Any severe CDI, no. (%)			
IDSA/SHEA criteria severe	24 (46.2)	23 (76.7)	<0.01
MN criteria mild	5 (9.6)	0 (0)	0.0800
MN criteria moderate	35 (67.3)	13 (43.3)	0.0340
MN criteria severe	10 (19.2)	15 (50.0)	<0.01
MN criteria super severe	2 (3.8)	2 (6.7)	0.5680
Laboratory data			
White blood cell count > 15,000/mm^3^, no. (%)	11 (21.2)	8 (26.7)	0.5690
Albumin < 3, no. (%)	37 (71.2)	25 (83.3)	0.2160
Creatinine > 1.5 mg/dL, no. (%)	14 (26.9)	20 (66.7)	<0.01
C reactive protein ≥ 10 mg/dL, no. (%)	11 (21.2)	9 (30.0)	0.3690
C reactive protein ≥ 15 mg/dL, no. (%)	7 (13.5)	5 (16.7)	0.6920
Antibiotics administration at CDI onset, no. (%)			
Penicillin	2 (3.8)	3 (10.0)	0.2620
Cephalosporin	27 (51.9)	16 (53.3)	0.9020
Carbapenem	12 (23.1)	5 (16.7)	0.4904
Fluoroquinolone	8 (15.4)	1 (3.3)	0.0930
Clindamycin	0 (0)	0 (0)	-
β-lactam/β-lactamase inhibitor	20 (38.5)	8 (26.7)	0.2780
Anti-viral agents	0 (0)	1 (3.3)	0.1853
Anti-fungal agents	0 (0)	2 (6.7)	0.0594
None	10 (19.2)	9 (30.0)	0.2660
Concomitant medication use, no. (%)			
PPIs	17 (32.7)	15 (50.0)	0.1220
H_2_RAs	2 (3.8)	4 (13.3)	0.1120
P-CAB	16 (30.8)	4 (13.3)	0.0770
Steroid	12 (23.1)	7 (23.3)	0.9790
Immunosuppressants other than steroids	15 (28.8)	10 (33.3)	0.6707
Anti-cancer chemotherapy	7 (13.5)	3 (10.0)	0.6450
Risk factors for recurrence of CDI			
Age ≥ 65 years, no. (%)	34 (65.4)	25 (83.3)	0.08
Use of antimicrobials for at least 3 days after CDI diagnosis, no. (%)	23 (44.2)	11 (36.7)	0.50
Chronic kidney disease, no. (%)	19 (36.5)	23 (76.7)	<0.01
Past medical history of CDI, no. (%)	4 (7.7)	3 (10.0)	0.71
Use of proton pump inhibitors, no. (%)	17 (32.7)	15 (50.0)	0.12
Use of H_2_ blocker, no. (%)	2 (3.8)	4 (13.3)	0.11
Use of potassium-competitive acid blocker, no. (%)	16 (30.8)	4 (13.3)	0.07
Past hospitalization within 3 months before onset of CDI, no. (%)	24 (46.2)	9 (30.0)	0.15
Use of fluoroquinolone within one month of onset of CDI, no. (%)	8 (15.4)	1 (3.3)	0.09
Use of carbapenem within one month of onset of CDI, no. (%)	12 (23.1)	5 (16.7)	0.49
Solid malignancy, no. (%)	16 (30.8)	9 (30.0)	0.94
Hematologic cancer, no. (%)	3 (5.8)	3 (10.0)	0.48
ICU at diagnosis of CDI, no. (%)	3 (5.8)	2 (6.7)	0.87
History of cephalosporin use prior to CDI diagnosis, no. (%)	27 (51.9)	16 (53.3)	0.90
History of abdominal surgery, no. (%)	15 (28.8)	4 (13.3)	0.10
Using steroid, no. (%)	12 (23.1)	7 (23.3)	0.98
Enteral feeding, no. (%)	5 (9.6)	7 (23.3)	0.09
Inflammatory bowel disease, no. (%)	4 (7.7)	1 (3.3)	0.43
Severe CDI by IDSA/SHEA criteria, no. (%)	24 (46.2)	23 (76.7)	<0.01
Severe and super severe CDI by MN criteria, no. (%)	12 (23.1)	17 (56.7)	<0.01
Number of risk factors of CDI recurrence, median (range)	5 (2–11)	6 (3–10)	0.2720
Probiotics before CDI diagnosis, no. (%)	15 (28.8)	14 (46.7)	0.1040
Probiotics after CDI diagnosis, no. (%)	36 (69.2)	23 (76.7)	0.4700
Global cure	34 (65.4)	26 (86.7)	<0.05
Clinical cure	36 (69.2)	27 (90.0)	<0.05
Recurrence	6 (11.5)	1 (3.3)	0.2000
Adverse effect, no. (%)	0 (0)	0 (0)	-
All-cause 30-day mortality after CDI treatment	2 (3.8)	0 (0)	0.2770

IQR, interquartile range; CDI, Clostridioides difficile infection; IDSA/SHEA, Society for Healthcare Epidemiology of America/Infectious Diseases Society of America; P-CAB, potassium-competitive acid blocker; PPI, proton pump inhibitor; H_2_RAs, histamin-2 receptor antagonist.

**Table 2 antibiotics-14-00228-t002:** Multivariate logistic regression analysis for variables associated with clinical cure.

Variable	Clinical Cure (*n* = 63)	Clinical Failure (*n* = 19)	*p* Value	Adjusted Odds Ratio (95% CI)	*p* Value
Chronic kidney disease	35 (55.6%)	7 (36.8%)	0.121		
Combination with BEZ	27 (42.9%)	3 (15.8%)	0.027	4.167 (1.029–16.885)	0.046
Probiotics after CDI diagnosis	49 (77.8%)	10 (52.6%)	0.035	3.403 (1.045–11.079)	0.042
Steroid use	11 (17.5%)	8 (42.1%)	0.031	0.236 (0.068–0.815)	0.022

BEZ, bezlotoxumab; CDI, *Clostridioides difficile* infection.

**Table 3 antibiotics-14-00228-t003:** Clinical characteristics of the only severe cases defined by the IDSA/SHEA guidelines for CDI.

Variables	FDX (*n* = 24)	FDX + BEZ (*n* = 23)	*p* Value
Age (years), median (IQR)	76 (44–86)	76 (36–90)	0.6480
Age ≥ 65 years, no. (%)	18 (75.0)	19 (82.6)	0.5240
Age ≥ 75 years, no. (%)	14 (58.3)	13 (56.5)	0.9000
Age ≥ 80 years, no. (%)	7 (29.1)	9 (39.1)	0.4710
Female sex, no. (%)	8 (33.3)	11 (47.8)	0.3120
Body mass index < 18.5, no. (%)	9 (37.5)	7 (30.4)	0.6090
Past medical history of CDI, no. (%)	2 (8.3)	1 (4.3)	0.5760
Past hospitalization within 3 months before onset of CDI, no. (%)	13 (54.1)	6 (26.0)	0.0500
Past hospitalization < 1 year, no. (%)	18 (75.0)	10 (43.4)	0.028
Temperature > 37.8 °C at diagnosis of CDI, no. (%)	11 (45.8)	10 (43.4)	0.8710
Abdominal pain at diagnosis of CDI, no. (%)	11 (45.8)	12 (52.1)	0.4270
Bowel movements > 5 at diagnosis of CDI, no. (%)	11 (45.8)	11 (47.8)	0.6640
Bloody stool, no. (%)	2 (8.3)	1 (4.3)	0.5760
Fasting, no. (%)	5 (20.8)	2 (8.6)	0.2430
Enteral feeding, no. (%)	1 (4.1)	7 (30.4)	0.0170
Hospitalization for more than 1 month before onset of CDI, no. (%)	11 (45.8)	9 (39.1)	0.6420
Use of antimicrobials for at least 3 days after CDI diagnosis, no. (%)	13 (54.1)	8 (34.7)	0.1810
Comorbidities, no. (%)			
Diabetes mellitus	13 (54.1)	10 (43.4)	0.4640
Chronic kidney disease	16 (66.6)	20 (86.9)	0.1010
HD	10 (41.6)	3 (13.0)	0.0280
Heart failure/ischemic heart disease	7 (29.1)	9 (39.1)	0.4710
Chronic liver disease	2 (8.3)	0 (0)	0.1570
Chronic obstructive pulmonary disease	1 (4.1)	2 (8.6)	0.5250
Cerebrovascular disease	2 (8.3)	5 (21.7)	0.1970
Inflammatory bowel disease	1 (4.1)	0 (0)	0.3220
Solid malignancy	6 (25.0)	8 (34.7)	0.4640
Hematologic malignancy	3 (12.5)	2 (8.6)	0.6720
History of abdominal surgery, no. (%)	7 (29.1)	3 (13.0)	0.1770
Community onset CDI, no. (%)	3 (12.5)	2 (8.6)	0.3430
ICU at diagnosis of CDI, no. (%)	1 (4.1)	2 (8.6)	0.5250
Any severe CDI, no. (%)			
MN criteria mild	0 (0)	0 (0)	-
MN criteria moderate	15 (62.5)	9 (39.1)	0.1090
MN criteria severe	7 (29.1)	12 (52.1)	0.1080
MN criteria super severe	2 (8.3)	2 (8.6)	0.9650
Laboratory data			
White blood cell count > 15,000/mm^3^, no. (%)	11 (45.8)	8 (34.7)	0.4400
Albumin < 3, no. (%)	16 (66.6)	20 (86.9)	0.1010
Creatinine > 1.5 mg/dL, no. (%)	14 (58.3)	20 (86.9)	0.0280
C reactive protein ≥ 10 mg/dL, no. (%)	5 (20.8)	8 (34.7)	0.2850
C reactive protein ≥ 15 mg/dL, no. (%)	3 (12.5)	4 (17.3)	0.6380
Antibiotics administration at CDI onset, no. (%)			
Penicillin	1 (4.1)	3 (13.0)	0.2760
Cephalosporin	12 (50.0)	14 (60.8)	0.4540
Carbapenem	7 (29.1)	4 (17.3)	0.3410
Fluoroquinolone	2 (8.3)	1 (4.3)	0.5760
Clindamycin	0 (0)	0 (0)	-
β-lactam/β-lactamase inhibitor	7 (29.1)	6 (26.0)	0.8130
Anti-viral agents	0 (0)	1 (4.3)	0.3020
Anti-fungal agents	0 (0)	2 (8.6)	0.1400
None	2 (8.3)	6 (26.0)	0.1050
Concomitant medication use, no. (%)			
PPIs	7 (29.1)	13 (56.5)	0.0580
H_2_RAs	0 (0)	2 (8.6)	0.1400
P-CAB	10 (41.6)	3 (13.0)	0.0280
Steroid	8 (33.3)	3 (13.0)	0.1010
Immunosuppressants other than steroids	8 (33.3)	5 (21.7)	0.3740
Anti-cancer chemotherapy	2 (8.3)	2 (8.6)	0.9650
Probiotics before CDI diagnosis, no. (%)	10 (41.6)	12 (52.1)	0.4710
Probiotics after CDI diagnosis, no. (%)	18 (75.0)	17 (73.9)	0.9320
Global cure	15 (62.5)	21 (91.3)	<0.05
Clinical cure	15 (62.5)	22 (95.6)	<0.01
Recurrence	2 (8.3)	1 (4.3)	0.5760
Adverse effect, no. (%)	0 (0)	0 (0)	-
All-cause 30-day mortality after CDI treatment	2 (8.3)	0 (0)	0.1570

## Data Availability

Data are contained within the article.

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
