# Peer review of "Evaluating Bezlotoxumab-Fidaxomicin Combination Therapy in Clostridioides Infection: A Single-Center Retrospective Study from Aichi Prefecture, Japan"

_antibiotics, 2025, doi:10.3390/antibiotics14030228_

Round 1

Reviewer 1 Report

Comments and Suggestions for Authors

In the manuscript, a single-center clinical evaluation of the efficacy and safety of FDX+BEZ combination therapy in <100 CDI patients compared with FDX monotherapy was implemented. The clinical statistics demonstrated this seemingly promising treatment option for CDI patients with high risk of recurrence. Manuscript was generally well-organized with a clear conclusion and specified limits essentially due to the size of trials. 

MAb treatment is attractive option largely due to its both target specificity and ability to synergize with the host’s immune response. MAb therapy's exotoxin non-targeting nature also greatly facilitated its potential solution to AMR issue. That being said, difficulties persist on fully elucidating the mechanism of the synergic effect of mAb and classic antibiotics. Although not the focus of theis manuscript, the authors were encouraged to further proposed the MoA of FDX+BEZ, or suggested a synergistic heatmap to fundamentally discuss it with potential readers.

The authors also discussed the benefit of FDX over VAN in the combination therapy on maintaining the gut microbiota of patients - although it could be a secondary consideration consider the high mortality of CDI. 

Author Response

Comment:

The authors were encouraged to further propose the MoA of FDX+BEZ, or suggested a synergistic heatmap to fundamentally discuss it with potential readers.

Dear Response:

We sincerely appreciate your insightful comment regarding the mechanism of action (MoA) of FDX+BEZ and the suggestion to illustrate the synergistic effect using a heat map. While a comprehensive experimental analysis to elucidate the precise MoA of this combination therapy is beyond the scope of our current study, we have expanded our discussion of the potential synergy between FDX and BEZ. FDX exerts its antibacterial effect by inhibiting RNA polymerase, effectively reducing Clostridioides difficile bacterial load while preserving the gut microbiota. BEZ, a monoclonal antibody targeting toxin B, prevents its cytotoxic effect on intestinal epithelial cells, thereby reducing inflammation and preventing recurrence. The combination therapy is thought to work synergistically by simultaneously reducing bacterial load and neutralizing the toxins that drive mucosal damage and disease recurrence. We have now included this expanded discussion in the revised manuscript. While we recognize the value of a synergistic heat map, we have chosen to focus on discussing the known MoA of combination therapy based on the existing literature and our clinical findings. We hope that this addition strengthens the readers' mechanistic understanding of the FDX+BEZ synergy. Please check lines 156-170.

Reviewer 2 Report

Comments and Suggestions for Authors

The authors report clinical data for patients diagnosed with Clostridioides difficile infection (CDI) with one group treated with Fidaxomicin (FDX) and the other with FDX+ Bezlotoxumab (BEZ). Among the 82 patients, the FDX+BEZ group (n=30) demonstrated significantly higher cure rates compared with the FDX-alone group (n=52). The combination treatment data suggested the following:

  1. Improvements in the number of diarrhea cases.
  2. The recurrence rate in the FDX+BEZ group were lower compared to FDX-only treated group.
  3. Administration of BEZ was given at ~5.4 days after initiation of FDX. Although the study was not able to determine if timing of BEZ administration is critical, the authors report that such inference could be made only with larger datasets.

Overall, the manuscript provides a promise to FDX+BEZ combination treatment therapy for CDI patients with high risk of recurrence. The manuscript is well wirtten and covers various points to consider for future studies.

Author Response

Comment:
Administration of BEZ was given at ~5.4 days after initiation of FDX. Although the study was not able to determine if timing of BEZ administration is critical, the authors report that such inference could be made only with larger datasets.

Dear Response:
We appreciate your valuable comment regarding the timing of BEZ administration. To further explore this question, we conducted a subgroup analysis comparing clinical outcomes in patients who received BEZ within 5 days of FDX initiation (n = 13) versus those who received BEZ after 5 days (n = 17).

Our univariate analysis revealed no statistically significant differences between the two groups for:

  • Clinical cure (p = 0.390)
  • Global cure (p = 0.773)
  • Recurrence rate (p = 0.374)

These results suggest that, within this timeframe, the timing of BEZ administration may not have a major impact on treatment efficacy. However, given the limited sample size, we acknowledge that larger prospective studies are necessary to confirm these findings.

We have now incorporated these results into the revised manuscript. For conciseness, the detailed data are not shown in a separate table. The Discussion section has also been expanded to address the potential implications of BEZ timing in CDI management.

Manuscript Revision (Results & Discussion Sections)

Results (New Subgroup Analysis)

A subgroup analysis was conducted to assess the potential impact of BEZ administration timing on treatment outcomes. Patients who received BEZ within 5 days of FDX initiation (n = 13) were compared to those who received BEZ after 5 days (n = 17). No statistically significant differences were found between the two groups for Clinical cure (p = 0.390), Global cure (p = 0.773), or Recurrence rate (p = 0.374). For clarity and conciseness, detailed data are not shown but are available upon request. Please check lines 120-125.

Discussion (Implications of BEZ Timing)

The optimal timing of BEZ administration in CDI treatment remains a subject of clinical interest. In our study, we observed no significant difference in clinical outcomes based on whether BEZ was administered within or after 5 days of FDX initiation. Given our sample size limitations, larger prospective studies are required to definitively determine the optimal BEZ timing in CDI management. Please check lines 179-184.

Reviewer 3 Report

Comments and Suggestions for Authors

Clostridioides difficile infection (CDI) presents a significant healthcare challenge, which requires effective treatment strategies. In this study, Hirai et al. evaluated the combination therapy of bezlotoxumab and fidaxomicin using real-world clinical data, providing valuable insights through comprehensive data analysis. I recommend its publication, provided the following questions are better explained in the manuscript.

Minor Comments:

  1. The method of patient assignment to treatment groups is unclear. Was it a randomized study, or were treatment decisions based on clinical judgment? Please clarify.
  2. The manuscript states that the two groups were "well-matched," yet chronic kidney disease (CKD) was significantly different between them. Could this imbalance have impacted treatment outcomes? If so, how was it accounted for in the analysis?

Author Response

Comment 1:

"The method of patient assignment to treatment groups is unclear. Was it a randomized study, or were treatment decisions based on clinical judgment? Please clarify."

Dear Response:

We apologize for the lack of clarity. This was a retrospective study, and patient allocation to treatment groups was based on clinical judgment rather than randomization. Treatment decisions were made by attending physicians considering patients’ risk factors and severity of CDI. We have revised the Methods section to explicitly state this. Please check lines 145-147 and 265-268.

Comment 2:

"The manuscript states that the two groups were 'well-matched,' yet chronic kidney disease (CKD) was significantly different between them. Could this imbalance have impacted treatment outcomes? If so, how was it accounted for in the analysis?"

Dear Response:

Thank you for your insightful comment. You are correct that CKD was significantly more common in the FDX+BEZ group. However, when comparing patients who achieved clinical cure to those who failed, the prevalence of CKD was not significantly different (55.6% vs. 36.8%, p = 0.121). Instead, we observed significant differences between these groups in FDX+BEZ therapy (p = 0.027), probiotic use after CDI diagnosis (p = 0.035), and steroid use (p = 0.031). Given their potential influence on clinical cure, we included these three factors in a multivariable analysis. As shown in Table 2, FDX+BEZ therapy and probiotic use after CDI diagnosis were significantly associated with clinical cure, whereas steroid use was significantly associated with clinical cure failure. Please check Table 2.